# High risk injection drug use and uptake of HIV prevention and treatment services among people who inject drugs in KwaZulu-Natal, South Africa

Brian C. Zanoni[1,2,3]*, Cecilia Milford[4], Kedibone Sithole[4], Nzwakie Mosery[4], Michael Wilson[5,6], Shannon Bosman[7], Jennifer Smit[4]

1 Emory University, Atlanta, Georgia, United States of America, 2 Children's Healthcare of Atlanta, Atlanta, Georgia, United States of America, 3 Rollins School of Public Health, Atlanta, Georgia, United States of America, 4 Faculty of Health Sciences, MRU (MatCH Research Unit), School of Obstetrics and Gynecology, University of the Witwatersrand, Durban, South Africa, 5 Advance Access & Delivery, Durban, South Africa, 6 Department of Health Behaviour, University of North Carolina Gillings School of Global Public Health, Chapel Hill, NC, United States of America, 7 Centre for Community Based Research, Human Sciences Research Council, Pietermaritzburg, South Africa

* bzanoni@emory.edu

## Abstract

We conducted a mixed-methods study to understand current drug use practices and access to healthcare services for people who use injection drugs in KwaZulu-Natal, South Africa. We used respondent-driven sampling to recruit 45 people who used injection drugs within the past 6 months from KwaZulu-Natal, South Africa. We found high rates of practices that increase HIV/viral hepatitis risk including the use of shared needles (43%) and direct blood injections (bluetoothing) (18%). Despite 35% living with HIV, only 40% accessed antiretroviral therapy within the past year, and one accessed PrEP. None of the participants ever tested for Hepatitis C.

## Introduction

South Africa has an estimated 7.7 million individuals living with HIV, and KwaZulu-Natal has the highest HIV prevalence in the country [1]. In South Africa, there are more than 5 million people accessing antiretroviral therapy (ART), making it the largest national antiretroviral program in the world [1]. However, hidden populations remain that are not reached in the HIV prevention and care continuum, and efforts to expand HIV prevention, HIV testing and linkage to care services in sub-Saharan Africa have largely targeted non-injection drug using communities [1,2]. Before a recent increase in people who inject drugs (PWID) in South Africa, it was estimated that 21% of PWID in South Africa were living with HIV [1,3] compared to 14% in the general population [4]. A recent increase in injection drug use, coupled with poor access to addiction services in the forms of needle exchanges or medication assisted therapy (such as methadone or buprenorphine), has led to the increase in a sequestered population that has

**Data Availability Statement:** The data set has been uploaded to the Open Science Frameworks website and is available at: https://osf.io/6vfbz/.

**Funding:** This work was supported via Center for AIDS Research at Emory University (P30AI050409); PI: BCZ. The sponsors played no role in the study design, data collection, analysis or decision to publish.

**Competing interests:** The authors have declared that no competing interests exist.

potential to reverse many of the investments and gains in public health access to HIV prevention and treatment services [5]. The population of PWID in South Africa spans groups of homeless, sex workers, and working-class individuals, bridging multiple social networks contributing to the potential increase in HIV incidence among individuals not targeted in typical HIV continuum interventions [6].

Whoonga, an opiate-based street drug has been present in KwaZulu-Natal for more than 10 years, yet has only recently seen an increase in injection use [7,8]. In the past, Whoonga was a drug that was smoked; however, intravenous and subcutaneous administration has become more prevalent [9]. The recent increase in cases of infective endocarditis (infection of the heart typically from blood stream infections) in South Africa is indicative of an increase in injection drug use [5,7,8]. In addition, the dangerous practice of "bluetoothing" in which blood is withdrawn from one individual who has recently injected a drug and directly injected intravenously into another person, has been reported [10,11]. Called flash-blooding in other African countries, this practice is often performed in settings of poverty and poor access to needles or needle exchange programs [12–14]. This practice, in addition to needle sharing, has potential to increase the risk of HIV and viral hepatitis among PWID and spill over into the general population through sexual networks [15–18].

Given the limited data on the population of individuals using injection drugs in KwaZulu-Natal, South Africa, we conducted a mixed-methods, respondent-driven study to understand drug use practices and current access to health care in preparation for targeted prevention and treatment interventions.

## Methods

We used respondent driven sampling (RDS) to recruit individuals aged 18 years or older, with self-described use of injection drugs within the last 6 months, and currently living in KwaZulu-Natal, South Africa. We excluded individuals who did not speak either English or isiZulu, who were severely or visibly intoxicated, or those with severe mental or physical illness preventing participation in informed consent procedures.

We recruited three initial seed individuals who were attending a harm reduction center in Durban, South Africa. These three seed individuals were encouraged to recruit up to three other PWID from their individual social network. Each additional participant was also asked to recruit up to three different individuals from their own social network, until we reached a total sample of 45 participants in this pilot study. Recruitment and participation in interviews took place from November 1, 2021 to February 8, 2022. After providing written informed consent, participants completed a facilitated questionnaire that collected information on sociodemographics, drug use, sexual behavior and network characteristics, HIV testing practices, and use of HIV prevention or treatment services. Interviewers entered data directly into a REDCap database as questions were answered [19].

We assessed the frequency, type, and methods of drug use, information on needle procurement, use, and sharing using the National HIV Behavior Surveillance System; [20] and WHO ASSIST [21]. HIV risk assessment including sex work, number of sexual partners, condom use and frequency of condom use within the past 3 months was assessed by the Texas Christian University HIV/Hepatitis Risk Assessment [22]. We also evaluated access to health care by exploring knowledge of HIV status, HIV testing in the past 12 months, access and use of medical services in the past 12 months, knowledge and acceptability of HIV self-testing.

For this descriptive analysis we used standard summary statistics (e.g. counts/percentages; median and interquartile range of continuous measures). Basic descriptive data analyses were performed using REDCap. Results of in-depth interviews are reported separately.

**Table 1. Descriptive statistics of individuals recruited through respondent-driven sampling and self-reporting recent injection drug use in KwaZulu-Natal, South Africa.**

| Characteristic | Participants n (%) N = 45 |
|---|---|
| **Demographics** | |
| Median Age (years)(IQR) | 28.5 (26.6–32.3) |
| Male | 26 (58%) |
| Heterosexual | 41 (91%) |
| Single | 17 (39%) |
| Long term partner but not married and not living together | 26 (59%) |
| Several casual partners | 2 (5%) |
| South African citizen | 45 (100%) |
| **Education** | |
| Primary education | 4 (9%) |
| Secondary education | 39 (87%) |
| Tertiary level | 2 (4%) |
| Median grade completed (IQR) | 10 (10–11) |
| **Employment** | |
| Full time employed | 1 (2%) |
| Employed part time | 11 (24%) |
| Unemployed | 17 (38%) |
| **Housing** | |
| Homeless in the past 12 months | 42 (93%) |
| Currently homeless | 32 (76%) |
| **Alcohol Use** | |
| No alcohol in the past year | 29 (64%) |
| Binge drinking more than once a month | 10 (22%) |
| **Injection drug use** | |
| Used injection drugs | 45 (100%) |
| Median age at first use (years) (IQR) | 22 (19–26) |
| Last used injection drugs in the past month | 42 (93%) |
| Injecting more than once a day | 41 (91%) |
| Injecting daily | 2 (4%) |
| Injecting more than once a week | 2 (4%) |
| Using Whoonga/heroin daily or more | 45 (100%) |
| Had an opioid overdose (in last year) | 10 (22%) |
| Median number of opioid overdoses (IQR) (in last year) | 3.5 (2–4.75) |
| Know of others who experienced lethal opioid overdose (in last year) | 29 (64%) |
| Median overdose deaths known about (in last year) | 2 (1–3) |
| Know of others with non-lethal opioid overdose (in last year) | 28 (62%) |
| **Most common place acquired needles** | |
| Needle exchange or NGO | 29 (64%) |
| Friend / acquaintance | 4 (11%) |
| Pharmacy | 1 (4%) |
| **Dispose of needle** | |
| Needle exchange or NGO | 25 (56%) |
| On the street | 8 (8%) |
| Trash | 7 (16%) |
| **High risk Injection Drug Use** | |
| New / unused Needle use in last 12 months | |

*(Continued)*

**Table 1.** (Continued)

| Characteristic | Participants n (%) N = 45 |
|---|---|
| Always | 14 (31%) |
| Sometimes | 22 (49%) |
| Rarely | 9 (20%) |
| Never | 0 (0%) |
| Median number of people shared needles with in last 12 months (IQR) | 0 (0–3) |
| Median number of people shared other drug-use material with in last 12 months (IQR) | 3 (0–5) |
| Median number of people shared drugs with in the last 12 months (IQR) | 3 (1–4) |
| Used a shared needle in the past 12 months | 19 (42%) |
| Used shared material in last 12 months | 33 (73%) |
| Bluetoothing in the last 12 months | 8 (18%) |
| Re-used needle from someone else at last injection | 10 (22%) |
| Bluetoothing at last injection | 2 (4%) |
| Did not dispose of needle at last injection | 29 (64%) |
| Injected with known HIV+ individual | 6 (13%) |
| Other drug use in last year | 24 (53%) |
| **Healthcare Services** | |
| Participated in a drug treatment program | 27 (60%) |
| Ever tested for HIV | 43 (96%) |
| Known to be living with HIV | 15 (35%) |
| Accessing ART in the last 12 months | 6 (40%) |
| Median number of HIV tests in last 2 years | 3 (1–8.5) |
| Never tested for Hepatitis C | 45 (100%) |
| Currently taking PrEP | 1 (2%) |
| Interested in PrEP | 24 (53%) |

This study was approved by the Institutional review Boards of Emory University and the University of Witwatersrand and the KwaZulu-Natal National Department of Health.

## Results

We interviewed 45 individuals in Durban, South Africa who reported recent injection drug use. Participants had a median age of 28.5 years (IQR 26.6–32.3) and were predominantly male (58%), heterosexual (91%), had completed secondary education (87%) and were currently (76%) or recently (92%) homeless as indicated by **Table 1**. The median reported starting age of injecting drug use was 22 years (IQR 19–26). The majority of participants did not use other (non-injection) drugs (85%) or alcohol (64%). Most participants (93%) reported use of injection drugs within the last month with 91% reporting averaging more than one injection per day. All participants (100%) reported daily use of Whoonga. Opioid overdoses were personally experienced in 22% of participants in the last year, with a median of 3.5 (IQR 2–4.75) individual episodes of overdosing. All participants (100%) reported ever re-using needles or equipment with 42% reporting use of shared needles in the past year and 73% reported sharing other drug preparation or use materials. Bluetoothing was practiced in 18% of individuals. Thirty-five percent of individuals were known to be living with HIV but only six (40%) reported accessing antiretroviral therapy within the past 12 months and one individual was taking pre-exposure prophylaxis (PrEP). None of the participants had ever tested for Hepatitis C.

## Discussion

In this study we found high rates of unsafe injection drug practices that included sharing and re-using needles and drug preparation materials as well as the practice of directly injecting blood from an individual who had recently injected (bluetoothing). These practices coupled with the low uptake of treatment and preventative services described in this study, including HIV testing, PrEP and ART services, could negate some of the efforts made in the HIV prevention and treatment continuum of care in South Africa.

The practice of bluetoothing has previously been described in South Africa [23] and has been termed flashblooding in Tanzania [12–14,24]. Although practiced in a minority of injection drug users (10% in Tanzania [24] and 18% in our sample), as an extreme version of needle sharing, this practice amplifies the risk of HIV and/or Hepatitis C acquisition. It has also only been described in areas of poverty and limited access to needle exchanges. However, screening for this practice is uncommon and should be incorporated into routine interviews with PWID to reduce stigma and allow for targeted harm reduction services.

The current and substantial efforts to combat the HIV epidemic in South Africa do not adequately address injection drug use. Although efforts are increasing to routinely screen for alcohol use in HIV care across South Africa [25], screening for drug use (and injection drug use specifically) is not routinely conducted, and drug treatment is rarely integrated with HIV care services. In addition, access to harm reduction services that included access to housing, food, and naloxone nasal spray may improve survival and adherence to medical therapies [12]. Outreach and mobile medical services targeting the homeless and drug using population could assist in introducing this population to medical services; however, evidence-based interventions for low-to-middle income countries (LMIC) are limited [26–28].

While the COVID-19 pandemic slowed global access to harm reduction and health services to people who use drugs, parts of South Africa saw a positive shift in care for PWID. Traditionally governmental and policing policies were antagonistic toward individuals with drug use disorder; [29] however, some areas began to support the provision of evidence-based harm reduction practices, especially to low-income and homeless people who use drugs. Nationally, needle exchange programs, harm reduction services, or medication assisted treatment are often difficult to access either due to their high-priced fee-for-service models or limited availability [30]. Although methadone syrup and sublingual buprenorphine are available and on the Department of Health Essential Medicine List for adult hospitals in South Africa, they are only available by prescription from clinicians through drug dependency programs. KwaZulu-Natal's only needle and syringe exchange program in Durban was reinstated during COVID-19 lockdown, and a harm reduction center was opened to provide daily opiate substitution therapy to more than 200 PWID. The limited resources available for harm reduction services have historically resulted in law enforcement and punishment of substance use as the predominant response to opiate addiction [31]. However after the COVID-19 lockdowns, Durban also saw a dramatic shift in the way that members of law enforcement engaged with people who use drugs, shifting the narrative of police as punitive to one of protectors and advocates of health services for PWID [32,33]. Continued work is needed to reduce ongoing stigma associated with PWID which limits access to HIV prevention services and harm reduction programs [34,35].

Without the availability of widespread interventions, injection drug use has continued to increase in South Africa with a prevalence of 0.6% in a population-based survey in 2012 with 0.2% reporting Whoonga use [36]. More recent studies estimate that up to 15% of South African youth engage in drug use [16,37]. However, small pilot studies have shown high retention rates, opiate reduction, and improvement in mental health in individuals accessing methadone

treatment in South Africa [38]. Yet, the lack of hepatitis C testing and access to PrEP indicate significant gaps in service delivery. Integration of harm reduction services into routine health-care could impact the opiate and HIV epidemics in South Africa.

The major limitations of this study include the small sample size and descriptive nature of this preliminary study. In addition, this study was conducted during COVID-19 when services were interrupted by various lockdown policies. There was also an influx of new people into the city who may not have been familiar with available services, and there were unsafe drug use practices happening as a result of disruption of needle and other supplies during lockdown. However, we are encouraged by the ability to recruit a difficult to reach population through respondent-driven sampling over a short period of time (3 months) and using only 3 initial seed individuals. In addition, since this study was performed with a small sample obtained by respondent-driven sampling from seeds accessing harm reduction services, the results cannot be generalized to all of South Africa or all of KwaZulu-Natal.

## Conclusion

The increased prevalence of injection drug use in South Africa along with unsafe injection practices and low uptake of preventative and treatment services, in particular HIV related services, has potential to reverse the significant gains made in HIV prevention services.

## Author Contributions

**Conceptualization:** Brian C. Zanoni, Cecilia Milford, Shannon Bosman, Jennifer Smit.

**Data curation:** Brian C. Zanoni, Cecilia Milford, Kedibone Sithole, Nzwakie Mosery, Michael Wilson, Shannon Bosman, Jennifer Smit.

**Formal analysis:** Brian C. Zanoni, Cecilia Milford, Michael Wilson, Shannon Bosman, Jennifer Smit.

**Funding acquisition:** Brian C. Zanoni.

**Investigation:** Brian C. Zanoni, Cecilia Milford, Kedibone Sithole, Nzwakie Mosery, Michael Wilson, Shannon Bosman, Jennifer Smit.

**Methodology:** Brian C. Zanoni, Cecilia Milford, Kedibone Sithole, Nzwakie Mosery, Michael Wilson, Jennifer Smit.

**Project administration:** Brian C. Zanoni, Cecilia Milford, Kedibone Sithole, Nzwakie Mosery, Michael Wilson, Jennifer Smit.

**Resources:** Brian C. Zanoni, Michael Wilson, Jennifer Smit.

**Software:** Jennifer Smit.

**Supervision:** Brian C. Zanoni, Cecilia Milford, Kedibone Sithole, Nzwakie Mosery, Michael Wilson, Jennifer Smit.

**Validation:** Cecilia Milford, Jennifer Smit.

**Visualization:** Shannon Bosman.

**Writing – original draft:** Brian C. Zanoni, Cecilia Milford, Kedibone Sithole, Nzwakie Mosery, Michael Wilson, Shannon Bosman, Jennifer Smit.

**Writing – review & editing:** Brian C. Zanoni, Cecilia Milford, Kedibone Sithole, Nzwakie Mosery, Michael Wilson, Shannon Bosman, Jennifer Smit.

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
