## [Decision Letter · Decision Letter 0]

20 Mar 2023

PONE-D-23-01161High risk injection drug use and uptake of HIV prevention and treatment services among people who inject drugs in KwaZulu-Natal, South AfricaPLOS ONE

Dear Dr. Zanoni,

Thank you for submitting your manuscript to PLOS ONE. After careful consideration, we feel that it has merit but does not fully meet PLOS ONE’s publication criteria as it currently stands. Therefore, we invite you to submit a revised version of the manuscript that addresses the points raised during the review process.  Thank you for submitting your paper to PLOS ONE. The two reviewers agreed that your manuscript has significant merit and addresses an important and understudied area.  They concurred that there were areas that could be strengthened in a major revision to the existing manuscript. If you are able to address these concerns in your manuscript revision and responses to reviewers, I believe PLOS ONE will be able to publish this manuscript in the future. Specifically addressing the questions of novel interventions and clarifying the methods and limitations of the snowball sampling approach for readers who are less familiar. Thank you for the important work you and your colleagues are doing. 

We look forward to receiving your revised manuscript.

Kind regards,

Benjamin Bearnot, M.D., M.P.H.

Academic Editor

PLOS ONE

Journal Requirements:

Reviewers' comments:

Reviewer's Responses to Questions

**Comments to the Author**

1. Is the manuscript technically sound, and do the data support the conclusions?

Reviewer #1: Yes

Reviewer #2: Partly

2. Has the statistical analysis been performed appropriately and rigorously? 

Reviewer #1: Yes

Reviewer #2: No

3. Have the authors made all data underlying the findings in their manuscript fully available?

Reviewer #1: Yes

Reviewer #2: Yes

4. Is the manuscript presented in an intelligible fashion and written in standard English?

Reviewer #1: Yes

Reviewer #2: Yes

5. Review Comments to the Author

Reviewer #1: PONE-D-23-01161

This is a mixed-methods study (n=45) to understand current drug use practices and access to HIV prevention and treatment services for PWID in KwaZuluNatal, South Africa. The authors used respondent-driven sampling to identify a population at high risk for HIV/viral hepatitis and disconnection from HIV services. The eligible population was individuals who reported injecting opiates within the past 6 months from Durban, KwaZulu-Natal, South Africa. They found high rates of practices that increase HIV/viral hepatitis risk. The author include local unsafe injection practices such as “flashblooding” or “bluetoothing” that could threaten the gains made in HIV prevention and treatment. Of the 35% of participants living with HIV in the sample, only 40% accessed antiretroviral therapy within the past year. There was no prior hepatitis C testing by self report. The importance of this study is access to a high risk population for HIV/hepatitis C, which has been underreported in the literature and has low utilization of HIV and hepatitis C prevention and treatment services. The paper would benefit from a greater emphasis on opportunities for novel program innovation to improve access of HIV/hepatitis prevention and treatment services. I have provided suggestions under major comments for potential avenues for innovation. Please also address how other resource limited countries address high risk injection practices that could be applied to KZN. Overall the paper is well-written and by addressing comments below, should merit publication.

Major comments

Is there any further published literature on bluetoothing / flashblooding or other similar practices in other countries and how the authors’ study findings compare? Please add this to the discussion to set the context of the significance of findings.

The paper could benefit from emphasizing opportunities for high impact novel interventions, in paragraph 2 of the Discussion. For example, the sample noted high rates of homeless and high rates of overdosing. Please address the need and feasibility to address, housing, food security, and Narcan availability, which may improve survival and adherence to harm reduction and HIV/hep C treatment programs? How can HIV/hep C programs be adapted to find this population that has high rates of homelessness.

Furthermore, how do the findings necessitate more locally tailored risk assessments to accurately identify high risk local practices. Are injection drug use risk screening tools adequate to account for practices that include bluetoothing? For example, Dawn K. Smith, Jeffrey H. Herbst et al. doi: 10.1097/ADM.0000000000000123, albeit a U.S. based sample, does not account for local variations in practice. How would more direct solicitation of these practices in one on one interactions reduce stigma and allow consumers of harm reduction services to be more forthcoming of their use?

Please also address how other resource limited countries are addressing high risk injection practices that could be applied to KZN.

Minor comments

Line 24—briefly define endocarditis for the non clinical audience

Results Lines 71 and 72 are a duplication from the methods. Please move the dates of interviews up to the methods Lines 39-41 where the authors report the same information and delete this sentence from the results.

Discussion Line 112-6 -rewrite for better clarity

In addition to the sentence on Line 116, “Nationally, needle exchange programs, harm reduction services, and 117 medication assisted treatment are often difficult to access either due to their high-priced, fee-for118 service models or limited availability. (24)” Can you mention the restrictions on providers to medication assisted treatment /buprenorphine (e.g. are there special restrictions, licenses,/fees required for health care providers to deliver that restricts its implementation?).

Reviewer #2: This is a clearly written, brief report of a mixed methods study of 45 PWIDs in KwaZulu-Natal (KZN). The participants were recruited through respondent driven sampling (RDS) by starting with 3 individuals from a harm reduction center in Durban. It’s commendable that the authors were able to conduct the study during the omicron surge. The quantitative description of characteristics of the 45 individuals, however, resembles the preliminary report of a pilot study that will lay the foundation for a larger project. Of note, the qualitative results are not included in this paper.

The study understandably excluded those who were severely intoxicated or mentally incapacitated because of the need for informed consent but the authors do not provide a consort diagram or other data to indicate how well the 45 participants represent the pool of potential participants. The study surprisingly did not gather information on or otherwise describe the mental health status of the 45 ppts. All of these limitations make it hard to know whether it is reasonable to generalize from these findings to other PWIDs in the KZN region, let along other regions of SA or southern Africa. Perhaps this paper is more suited to a South African scientific journal.

Methods:

There is no justification given for choosing 45 as the size of the study group.

Would be helpful to see a justification for the choice of tools used for the questionnaire; two of the tools have often been used in US settings.

Hard to know to what extent are PWIDs accessing harm reduction programs in KZN when the RDS draws from social networks anchored in a harm reduction program.

Discussion: The main rationale for the study is that injection drug use is on the rise in SA but the citation for this is from ten years ago so is less persuasive ("Without the availability of widespread interventions, injection drug use has continued to increase in South Africa with a prevalence of 0.6% in a population based survey in 2012 with 0.2% reporting Whoonga use. (30)")

6. PLOS authors have the option to publish the peer review history of their article (what does this mean?). If published, this will include your full peer review and any attached files.

Reviewer #1: No

Reviewer #2: No

---

## [Author Response · Author response to Decision Letter 0]

3 Apr 2023

Response to reviewers

RE: High risk injection drug use and uptake of HIV prevention and treatment services among people who inject drugs in KwaZulu-Natal, South Africa– PONE-D-23-01161

April 3, 2023

Dear Journal of PLoS One editorial staff:

We would like to thank the reviewers and editor for their helpful suggestions and comments regarding the manuscripts. We have revised the manuscript and addressed the reviewer concerns as highlighted below and included track changes in the original document. Changes are noted in line numbers from the final clean manuscript. We hope you will find this improved version of the manuscript acceptable for publication in PLoS One. Thank you for your time and consideration.

Reviewer 1:

Major comments:

1. Is there any further published literature on bluetoothing / flashblooding or other similar practices in other countries and how the authors’ study findings compare? Please add this to the discussion to set the context of the significance of findings.

RESPONSE: The published literature on this practice is sparce and why we are eager to publish our findings. We have now included an additional paragraph in the discussion on prior descriptions. Lines 84 – 88. 

“The practice of bluetoothing has previously been described in South Africa (1) and has been termed flashblooding in Tanzania.(2, 3, 4, 5) Although practiced in a minority of injection drug users (10% in Tanzania (5) and 18% in our sample), as an extreme version of needle sharing, this practice amplifies the risk of HIV or Hepatitis C acquisition. It has also only been described in areas of poverty and limited access to needle exchanges.”

2. The paper could benefit from emphasizing opportunities for high impact novel interventions, in paragraph 2 of the Discussion. For example, the sample noted high rates of homeless and high rates of overdosing. Please address the need and feasibility to address, housing, food security, and Narcan availability, which may improve survival and adherence to harm reduction and HIV/hep C treatment programs? How can HIV/hep C programs be adapted to find this population that has high rates of homelessness.

RESPONSE: We thank the reviewer for this important point. Unfortunately, there is a lack of evidence-based interventions in LMIC for this population. We included reference for the few interventions in the literature. We have now included this gap in knowledge in our discussion. Lines 94 - 98. 

“In addition, access to harm reduction services that included access to housing, food, and naloxone nasal spray may improve survival and adherence to medical therapies.(2) Outreach and mobile medical services targeting the homeless and drug using population could assist in introducing this population to medical services; however, evidence-based interventions for low-to-middle income (LMIC) countries are lacking. (26, 27, 28)”

3. Furthermore, how do the findings necessitate more locally tailored risk assessments to accurately identify high risk local practices. Are injection drug use risk screening tools adequate to account for practices that include bluetoothing? For example, Dawn K. Smith, Jeffrey H. Herbst et al. doi: 10.1097/ADM.0000000000000123, albeit a U.S. based sample, does not account for local variations in practice. How would more direct solicitation of these practices in one on one interactions reduce stigma and allow consumers of harm reduction services to be more forthcoming of their use?

RESPONSE: We agree with the reviewer that elicitation of this practice through typical screening questions is lacking. We have now included this in our discussion. Lines 88 – 89. 

“However, screening for this practice is uncommon and should be incorporated into routine interviews with PWID to reduce stigma and allow for targeted harm reduction services.”

4. Please also address how other resource limited countries are addressing high risk injection practices that could be applied to KZN. 

RESPONSE: There is very limited data on evidence-based interventions for PWID in LMIC. We highlight a small pilot study done in South Africa (lines 120 - 122); however other evidence is lacking.

“However, small pilot studies have shown high retention rates, opiate reduction, and improvement in mental health in individuals accessing methadone treatment in South Africa.(38)”

And lines: 96 – 98. 

“Outreach and mobile medical services targeting the homeless and drug using population could assist in introducing this population to medical services; however, evidence-based interventions for low-to-middle income countries (LMIC) are limited.(26, 27, 28)” 

Minor comments

5. Line 24—briefly define endocarditis for the non clinical audience

RESPONSE: We have included a definition of endocarditis: Lines 19 – 21. 

“The recent increase in cases of infective endocarditis (infection of the heart typically from blood stream infections) in South Africa is indicative of an increase in injection drug use.(7, 8, 9)”

6. Results Lines 71 and 72 are a duplication from the methods. Please move the dates of interviews up to the methods Lines 39-41 where the authors report the same information and delete this sentence from the results.

RESPONSE: The line was removed from the results section, and we added the dates to the methods section. Lines 41 – 42. 

“Recruitment and participation in interviews took place from November 1, 2021 to February 8, 2022.”

7. Discussion Line 112-6 -rewrite for better clarity

RESPONSE: We have rewritten this section. Lines 100 – 104. 

“While the COVID-19 pandemic slowed global access to harm reduction and health services to people who use drugs, parts of South Africa saw a positive shift in care for PWID. Traditionally governmental and policing policies were antagonistic toward individuals with drug use disorder; however, some areas began to support the provision of evidence-based harm reduction practices, especially to low-income and homeless people who use drugs.” 

8. In addition to the sentence on Line 116, “Nationally, needle exchange programs, harm reduction services, and 117 medication assisted treatment are often difficult to access either due to their high-priced, fee-for118 service models or limited availability. (24)” Can you mention the restrictions on providers to medication assisted treatment /buprenorphine (e.g. are there special restrictions, licenses,/fees required for health care providers to deliver that restricts its implementation?).

RESPONSE: We have included the availability of methadone and buprenorphine in South Africa in lines: 106 – 108. 

“Although methadone syrup and sublingual buprenorphine are available and on the Department of Health Essential Medicine List for adult hospitals in South Africa, they are only available by prescription from clinicians through drug dependency programs.”

Reviewer 2:

9. There is no justification given for choosing 45 as the size of the study group.

RESPONSE: This was a pilot study with limited funding which capped our enrollment at 45 participants. This is included in lines 39 – 41. 

“These three seed individuals were encouraged to recruit up to three other PWID from their individual social network. Each additional participant was also asked to recruit up to three different individuals from their own social network, until we reached a total sample of 45 participants in this pilot study.”

10. Would be helpful to see a justification for the choice of tools used for the questionnaire; two of the tools have often been used in US settings.

RESPONSE: We use the National HIV Behavior Surveillance System, WHO ASSIST and TCU HIV/Hepatitis Risk Assessment which are all validated tools assessing HIV risk and drug and alcohol use. The WHO ASSIST has been validated in South Africa. There are limited tools on injection drug use that have been validated in South Africa; therefore, we chose instruments that were validated in other settings for similar populations. 

11. Hard to know to what extent are PWIDs accessing harm reduction programs in KZN when the RDS draws from social networks anchored in a harm reduction program.

RESPONSE: We acknowledge this limitation and have added it to our limitations section. See lines 132 – 134. 

“In addition, since this study was performed with a small sample obtained by respondent-driven sampling from seeds accessing harm reduction services, the results cannot be generalized to all of South Africa or all of KwaZulu-Natal.

12. Discussion: The main rationale for the study is that injection drug use is on the rise in SA but the citation for this is from ten years ago so is less persuasive ("Without the availability of widespread interventions, injection drug use has continued to increase in South Africa with a prevalence of 0.6% in a population based survey in 2012 with 0.2% reporting Whoonga use. (30)")

RESPONSE: Published data on injection drug use is in South Africa is very limited. We have included an additional statement and included more recent references. Lines 120 – 122. 

“More recent studies estimate that up to 15% of South African youth engage in drug use.(16, 37)”

---

## [Decision Letter · Decision Letter 1]

13 Apr 2023

High risk injection drug use and uptake of HIV prevention and treatment services among people who inject drugs in KwaZulu-Natal, South Africa

PONE-D-23-01161R1

Dear Dr. Zanoni,

We’re pleased to inform you that your manuscript has been judged scientifically suitable for publication and will be formally accepted for publication once it meets all outstanding technical requirements.

Kind regards,

Benjamin Bearnot, M.D., M.P.H.

Academic Editor

PLOS ONE

Additional Editor Comments (optional):

Thank you for your careful attention to the reviewers' comments. Congratulations on the acceptance on this manuscript, and good luck continuing this important line of investigation!

Reviewers' comments:

Reviewer's Responses to Questions

**Comments to the Author**

1. If the authors have adequately addressed your comments raised in a previous round of review and you feel that this manuscript is now acceptable for publication, you may indicate that here to bypass the “Comments to the Author” section, enter your conflict of interest statement in the “Confidential to Editor” section, and submit your "Accept" recommendation.

Reviewer #1: All comments have been addressed

2. Is the manuscript technically sound, and do the data support the conclusions?

Reviewer #1: Yes

3. Has the statistical analysis been performed appropriately and rigorously? 

Reviewer #1: Yes

4. Have the authors made all data underlying the findings in their manuscript fully available?

Reviewer #1: Yes

5. Is the manuscript presented in an intelligible fashion and written in standard English?

Reviewer #1: Yes

6. Review Comments to the Author

Reviewer #1: The reviewer appreciates the receptivity of the authors to feedback and the authors have adequately addressed concerns in the response and in the manuscript. I have no further comments.

7. PLOS authors have the option to publish the peer review history of their article (what does this mean?). If published, this will include your full peer review and any attached files.

Reviewer #1: No

---

## [Editor Report · Acceptance letter]

5 May 2023

PONE-D-23-01161R1 

High risk injection drug use and uptake of HIV prevention and treatment services among people who inject drugs in KwaZulu-Natal, South Africa 

Dear Dr. Zanoni:

I'm pleased to inform you that your manuscript has been deemed suitable for publication in PLOS ONE. Congratulations! Your manuscript is now with our production department. 

Kind regards, 

on behalf of

Dr. Benjamin Bearnot 

Academic Editor

PLOS ONE